# Real-Life Experience on Directional Deep Brain Stimulation in Patients with Advanced Parkinson’s Disease

**DOI:** 10.3390/jpm12081224

**Published:** 2022-07-27

**Authors:** Maija Koivu, Filip Scheperjans, Johanna Eerola-Rautio, Nuutti Vartiainen, Julio Resendiz-Nieves, Riku Kivisaari, Eero Pekkonen

**Affiliations:** 1Department of Neurology, Helsinki University Hospital and Department of Clinical Neurosciences (Neurology), University of Helsinki, PL 00029 Helsinki, Finland; filip.scheperjans@hus.fi (F.S.); johanna.eerola-rautio@hus.fi (J.E.-R.); eero.pekkonen@hus.fi (E.P.); 2Department of Neurosurgery, Helsinki University Hospital, PL 00029 Helsinki, Finland; nuutti.vartiainen@hus.fi (N.V.); julio.resendiz-nieves@hus.fi (J.R.-N.); riku.kivisaari@hus.fi (R.K.)

**Keywords:** directional deep brain stimulation, single segment activation, motor symptoms, non-motor symptoms, advanced Parkinson’s disease

## Abstract

Directional deep brain stimulation (dDBS) is preferred by patients with advanced Parkinson’s disease (PD) and by programming neurologists. However, real-life data of dDBS use is still scarce. We reviewed the clinical data of 53 PD patients with dDBS to 18 months of follow-up. Directional stimulation was favored in 70.5% of dDBS leads, and single segment activation (SSA) was used in 60% of dDBS leads. Current with SSA was significantly lower than with other stimulation types. During the 6-month follow-up, a 44% improvement in the Unified Parkinson’s Disease Rating Scale (UPDRS-III) points and a 43% decline in the levodopa equivalent daily dosage (LEDD) was observed. After 18 months of follow-up, a 35% LEDD decrease was still noted. The Hoehn and Yahr (H&Y) stages and scores on item no 30 “postural stability” in UPDRS-III remained lower throughout the follow-up compared to baseline. Additionally, dDBS relieved non-motor symptoms during the 6 months of follow-up. Patients with bilateral SSA had similar clinical outcomes to those with other stimulation types. Directional stimulation appears to effectively reduce both motor and non-motor symptoms in advanced PD with minimal adverse effects in real-life clinical care.

## 1. Introduction

Early stage Parkinson’s disease (PD) is usually treated with oral dopaminergic medication. As the disease progresses, motor and non-motor symptoms and symptom fluctuations become more prominent despite optimal medication. This stage of the disease is considered the advanced stage, and device-aided therapies, such as intrajejunal levodopa infusion, subcutaneous apomorphine infusion, and deep brain stimulation (DBS), should be considered to alleviate symptoms and enhance the quality of life in patients with Parkinson’s disease [1]. DBS is generally considered for patients with well-preserved cognition and without a history of falls and balance difficulties [1].

DBS has become one of the main treatment options for patients with advanced PD [2,3]. For treating motor symptoms, the dorsolateral subthalamic nucleus (STN) is widely used as a stimulation target and, to some extent, also the globus pallidus interna (GPi) [4,5]. In PD, the loss of dopaminergic neurons in the substantia nigra correlates with PD symptom severity, in particular bradykinesia and rigidity [6,7]. With DBS treatment, these symptoms can be alleviated, and there are several proposed mechanisms of action [6]. Physiological effects include direct inhibition of neural activity caused by the activation of inhibitory axons, especially with high-frequency stimulation (>130 Hz), or direct excitation of neural activity. Other proposed mechanisms are “jamming” or information lesion, in which stimulation provides local transmission block, and synaptic filtering [6,7,8]. DBS might act as a filter, permitting modulation of the activated neurons [8]. DBS can also cause metabolic changes. High-frequency stimulation has been reported to increase the amount of dopamine receptor D1 receptors and decrease D2 and D3 receptors and enhance the release of neurotransmitters [6].

Since its introduction in the mid-2010s, directional deep brain stimulation (dDBS) has been preferred by PD patients and programming physicians, even though directional programming has been reported to be more time consuming [5,9,10]. Commercially available dDBS systems consist of an implantable pulse generator (IPG), coiled extensions, and two directional leads with electrodes in a 1-3-3-1 order in contrast to the traditional DBS system with electrodes arranged in a 1-1-1-1 manner [11]. In treating severe Parkinsonian and essential tremor symptoms, a wider therapeutic window (TW), increased thresholds for side effects, and lower threshold for therapeutic stimulation have been reported using directional stimulation compared with conventional ring-mode (=omnipolar) stimulation [12,13,14,15,16]. When compared to conventional DBS leads, directional leads may even have wider TWs while activated in omnipolar mode [17].

However, long-term real-life data on the use of directional stimulation is still scarce [17,18,19]. The aim of this study was to examine the real-life use of dDBS in advanced PD patients treated with STN dDBS.

## 2. Materials and Methods

The data was collected retrospectively from the Helsinki University Hospital medical records and Abbott Clinician Programmer (Abbott Neuromodulation, Austin, TX, USA) used by the programming neurologists in the clinic. The following demographic data was obtained: patients’ age, duration of PD and advanced PD, and other medical comorbidities, including the use of anticoagulants/antiplatelet drugs and antidepressants. For this study, the Hoehn and Yahr (H&Y) stage, the use of anti-Parkinsonian medication (calculated as the levodopa equivalent daily dosage, LEDD [20]) was collected at fixed time points: baseline (i.e., the screening visit for DBS eligibility), at the 1- and 6-month inpatient programming sessions, followed by outpatient visits at 12 and 18 months. To assess the clinical benefit of DBS stimulation, the changes in the Unified Parkinson’s Rating Scale part III (UPDRS-III), Abnormal Involuntary Movement Scale (AIMS), Parkinson Disease Questionnaire 39 (PDQ-39), and Non-Motor Symptoms Questionnaire (NMS-Quest) were evaluated. The UPDRS-III scores were available at baseline (medication OFF and medication ON states) and at the monopolar and directional survey at 1, 1.5, and 6 months (medication OFF, stimulation ON state). To assess possible DBS-related dysarthria, the score of the UPDRS-III item no 18 “speech” was separately reviewed. Item no 30 “postural stability” in the UPDRS-III scale was independently examined for possible DBS-related balance disorder. The scores of the Mini-Mental Status Examination (MMSE) and Beck Depression Inventory (BDI) at baseline and at the 6-month visit were gathered to evaluate possible DBS-related cognitive and/or mood problems.

The DBS settings (i.e., amplitude, pulse width, frequency, and impedance), the activation of directional stimulation, and the use of individual programs were examined. Single segment activation (SSA) was compared with two segment activation (TSA) and/or omnipolar stimulation. The consumption of the IPG battery was also analyzed with different stimulation types.

The patient selection for DBS treatment was assessed according to accepted guidelines [21]. Dorsolateral STN was the target in all implantations. DBS operations were performed initially under local anesthesia with intraoperative testing (18/63 operations) and later, in general anesthesia (45/63). On the first postoperative day, the dDBS system (Abbott Infinity DBS™, Abbott Neuromodulation, Austin, TX, USA) was turned on with a 50 percent reduction in the levodopa dosage. DBS programming was conducted in a routine protocol established in our clinic after about 12-h Parkinsonian drug withdrawal (medication OFF, stimulation ON state), adapted from a previously published proposal [22], as shown in Figure 1.

After the 6-month inpatient programming visits, the patients were transferred to our outpatient clinic, with follow-up visits at 12 and 18 months. During these visits, the effect of DBS was assessed with the medication ON and stimulation ON state.

Because of the retrospective registry assessment, this study was approved by the director responsible for the academic research at the Helsinki University Hospital without an additional approval from ethics committee, according to the Finnish laws. The statistical analysis was performed using SPSS Statistics 27 (IBM Corp, Armonk, NY, USA). The data is presented as the median (interquartile range, IQR) and mean (standard deviation, SD) when applicable. The distributions of the variables were evaluated with the Shapiro–Wilk test. Data analysis was conducted using the paired *t*-test, Wilcoxon signed rank test, and repeated measurements ANOVA and Friedman test for repeated measurements. Bonferroni correction was used to account for multiple comparisons. *p*-values < 0.05 were considered significant.

## 3. Results

In total, 63 patients were implanted with a subthalamic dDBS system, and 62/63 operations were bilateral implantations. In total, 45 DBS operations (72%) were performed under general anesthesia and 18 (28%) under local anesthesia. The use of DBS stimulation in 125 dDBS leads was examined up to 6 months. Due to the COVID-19 pandemic and patient dropout, data of 105 dDBS leads was available for the 18-month follow-up. 

Patients’ demographics are presented in Table 1. Clinical data from the baseline and six-month visit is presented in Table 2. When compared to baseline, UPDRS-III improved by a median of 15.0 points (IQR 13.0 points) corresponding to a median 44% improvement (IQR 31%). The AIMS and H&Y scores decreased significantly (*p* < 0.001 in both). Patients also reported less non-motor symptoms as measured with NMS-Quest, as shown in Table 2. The PDQ-SI score at 6 months vs. baseline did not change significantly (*p* = 0.652). When each domain was separately examined, improvement was noted in the scores of the following domains: communication, bodily discomfort, and stigma (*p* = 0.007–0.016).

After the initial 6-month programming, the H&Y stages were at a median of 2.0 (IQR 0.5) throughout the follow-up, and thus were lower than at baseline (*p* < 0.001 when 12- and 18-month scores were compared to baseline). Item no 30 “postural stability” in UPDRS-III dropped from a median of 1.0 (IQR 1.0) at baseline to a median of 0.0 (IQR 1.0) at the 6-month visit and remained stable until the end of the follow-up (*p* < 0.001 at all time points when compared to baseline).

After the DBS implantation, LEDD decreased and remained lower than at baseline throughout the follow-up (*p* < 0.001), as shown in Figure 2. LEDD reductions corresponded to a median 42.7% decrease (IQR 36.2%) at 6 months and to 35.4% (IQR 40.5%) at 18 months, even though a slight increase in LEDD was observed at the end of the follow-up compared to the 6-month LEDD (*p* = 0.013).

After the 6-month inpatient programming visits, 64.8% of dDBS leads had the same electrode configuration as that after the directional survey. The majority of the alterations in electrode activation was a shift from SSA to another SSA at the same electrode contact level (in 7.2% of the dDBS leads) or a switch from directionality to omnipolarity (8.0%) at the same contact level (Figure 3). Other alterations were switching from omnidirectional stimulation to directional stimulation (7.2%) and/or from TSA stimulation to SSA (5.6%). After the 6-month visit, SSA was used in 83/125 leads, omnipolarity only in 31/125, and TSA in 11/125 (Figure 4). During the 6-month in-hospital follow-up, 106/125 (84.8%) dDBS leads were programmed in directional mode. In 19/125 (15.2%) leads, in-hospital directionality testing was not performed due to sufficient symptom control with omnipolar stimulation.

In the majority of leads (87/105, 83%), the stimulation type remained the same from the 6-month up to end of the 18-month follow-up. In 18/105 (17%) dDBS leads, the stimulation type was changed from directional mode to omnidirectional mode (in 7.6% of dDBS leads), SSA was changed to TSA (2.8%), or SSA changed to another SSA within the same electrode level (Figure 3). The modifications were carried out to achieve better symptom control and avoid stimulation-induced side effects. In two patients, stimulation was changed in both dDBS leads. In one patient, SSA was changed to another SSA in one dDBS lead and SSA to omnipolarity in another lead. In another patient, SSA was changed to another SSA in both leads.

At the end of the 18-month follow-up, in dDBS leads with tested directionality, only 17/105 (16.2%) dDBS leads had omnipolar stimulation. This accounts for five patients with bilateral omnipolar stimulation (10/17 dDBS leads), and another seven patients had omnipolarity that was unilaterally active. After the follow-up, 70.5% of all dDBS had the directional mode active and only 29.5% had the omnidirectional mode. Mostly, directional stimulation was single segment activation, as shown in Figure 4.

The outcomes of patients with bilateral directional stimulation were compared to those patients with bilateral omnipolar stimulation and/or with unilateral directional stimulation and omnipolar stimulation. The changes in the UPDRS-III points and LEDD and the non-motor questionnaire or PDQ-39 scores did not differ between these groups (*p* = 0.127–0.201). Additionally, the baseline UPDRS-III scores, LEDD, and H&Y stage were similar and there were no differences in the gender or advanced disease duration (*p* = 0.288–0.927). The LEDD or UPDRS-III reductions were similar between those that had been operated on under general anesthesia and those with local anesthesia (*p* = 0.433–0.822).

Among all stimulation types included, current increased to a median of 0.9 mA (IQR 1.0 mA) during the first 6 months (*p* < 0.001) and a median of 0.2 mA (IQR 0.6 mA) until the 18-month visit (*p* = 0.001 vs. 6-month visit) (Table 3). The pulse width remained constant throughout the follow-up (median 60 µs, IQR 0.0 for all time points, *p*-values 0.092–0.688), with a range of 40–110 µs. The median frequency remained at 130 Hz and IQR was increased to 30.0, creating a statistical difference between baseline and the 6-, 12-, and 18-month visits (*p* = 0.001–0.002). The frequency ranged from 50 to 200 Hz individually. After the 6-month visit, the frequency at the 12-month or 18-month visit was similar to the frequency at the 6-month visit (*p* = 0.881–0.970). The median impedances rose significantly during the follow-up (*p* < 0.001 at all time points). When SSA was compared to TSA and/or omnipolar stimulation at the 6-, 12-, and 18-month visits, the current in SSA was significantly lower (*p* = 0.001–0.05) and the impedance was higher (*p* = 0.001–0.015) at all time points, but the frequencies or pulse widths did not differ (*p* = 0.144–0.720). In total, 98% of the patients had solely used one program, mostly directional, at home when examined at the 18-month visit. They presented various reasons for choosing directionality. When analyzed in this study, the patients’ UPDRS-III scores were lower with directional stimulation as compared to omnipolar stimulation (median 21.0, IQR 11 with directional stimulation vs. median 25.0, IQR 14.0 with omnipolar stimulation, *p* = 0.012).

The IPG battery voltage decreased from a median of 3.0 V (IQR 0.0 V) to a median of 2.95 V (IQR 0.02 V) within the first 12 months (*p* < 0.001 at 6 and 12 months vs. at implantation). It reduced to 2.84 volts (IQR 0.03) at the 18-month visit (*p* < 0.001 at 18 months vs. at implantation). At the 6-, 12-, and 18-month visits, there were no statistical differences in the battery consumption between those patients who had bilateral directional stimulation and those who had either unilateral directional stimulation and/or bilateral omnipolar stimulation (*p* = 0.553–0.926). There was no IPG change due to battery exhaustion during the 18-month follow up.

Only one dDBS lead was reimplanted within the 12 postoperative months because of an initial suboptimal placement. Because of another patient’s complaints of discomfort, an IPG was replaced subcutaneously. After a dDBS lead implantation, the manufacturer reported a lead defect in a dDBS lead that prevented further MRI imaging but did not require a lead replacement nor did it prevent the programming. Two patients (3.2%) had deep vein thrombosis within the first postoperative month that was treated with a brief anticoagulation treatment (one patient after DBS implantation with local anesthesia and one after general anesthesia). Four patients (6.3%) used a course of antibiotics for superficial wound infection. No severe dDBS system-related infections or operation-related hemorrhages were reported. One patient was admitted to the inpatient ward for an additional programming visit after the initial six-month inpatient programming schedule. The median score given for dysarthria in UPDRS-III was 0 points at baseline and 1 point at the 6-month visit, without a clear difference (*p* = 0.819).

## 4. Discussion

Most of the published directional stimulation studies have focused on the effect of directionality on TWs [12,13,14]. To our knowledge, this study is the first study of 18 months of real-world follow-up of 53 PD patients treated with subthalamic dDBS. In previous clinical studies on dDBS, follow-ups of 6–12 months have been reported, with a 48% LEDD reduction and an improvement of 15 points in UPDRS-III, yet with small patient cohorts (range 10–30 patients) [14,17,18,23]. An abstract on the 12-month outcomes of 129 patients with dDBS has been reported, with improvements in the UPDRS-III and PDQ-39 scores [19]. A longer follow-up of 15 months with dDBS-treated patients with PD and essential tremor was published with a focus on the use of the advanced features (directionality and/or shorter pulse width) of dDBS systems [9].

Mostly directional (SSA) stimulation was used throughout the follow-up in this study. After 18 months of follow-up, 29/53 patients used bilateral directional stimulation with SSA and only 14 patients had unilaterally SSA and another dDBS lead programmed either in omnipolar mode or directional stimulation with two segments active (TSA). Only nine patients had bilateral omnipolar stimulation. The current used in SSA was lower than in other stimulation types (*p* = 0.001–0.002) and the impedance was higher (*p* = 0.001–0.015). The frequencies or pulse widths did not differ between the stimulation types. The use of SSA in this study is analogous to previous reports [9,17,24]. Directional stimulation has been preferred in previous studies, and reports of 50–74% of patients receiving directional subthalamic stimulation, mostly SSA, have been published. With ventral intermediate thalamic DBS (VIM-DBS) stimulation, the number is similar (74–79%) [5,10,22,24]. In our study, patients preferred directional programs after comparing directional and omnidirectional programs in everyday life. This result corresponds to the result of the PROGRESS study in which 52.8% of subjects blinded to the stimulation type preferred directional stimulation [5]. Even when given the opportunity, most patients solely used one program, mostly directional, after the test period in our study. When asked, the patients reported various reasons for choosing directionality without any clear pattern of preference. However, when analyzed in this study, patients’ UPDRS-III scores were lower with directional stimulation compared to the UPDRS-III scores with omnipolar stimulation (*p* = 0.012). This suggests that in our patient cohort, patients probably had better symptom control with directionality. In the PROGRESS study, clinicians favored directionality for symptom relief in most patients (96/113) and to avoid side effects in 11/113 [5]. In another study with a small patient cohort (*n* = 33), similar results were reported [25].

The clinical outcomes in our study resemble prior results on alleviating Parkinsonian motor symptoms after dDBS implantation and are similar to those reported on conventional DBS treatment [14,21,23,26]. We observed approximately 44% improvement in UPDRS-III points and a 43% decline in LEDD after 6 months postoperative and a 35% decrease in LEDD at 18 months postoperative. The H&Y stages remained lower than at baseline throughout the follow-up. Non-motor symptoms, reported by the patients with NMS-Quest, were relieved with dDBS treatment. With the traditional DBS system, improvement of non-motor symptoms, especially sleep-, attention/memory-, and urinary-related symptoms, has been reported [27,28,29]. Pain-related symptoms, as evaluated with the PDQ-39′s domain “bodily discomfort”, were relieved with dDBS treatment in this study. In a recent meta-analysis, it was noted that STN- and GPi-DBS can relieve pain, with a 40% improvement in pain scores [30]. The pain alleviation can last up to 8 years, as observed in a follow-up study [31]. Interestingly, in our cohort, no comprehensive improvement in PDQ-SI was observed, yet the scores in the domains of the aforementioned bodily discomfort, communication, and stigma showed clear improvement as previously reported [5,19]. Some patients may exaggerate daily activity functions and underestimate motor symptoms prior to surgery and, therefore, misinterpret the improvement of motor symptoms by the DBS [32]. This effect may have influenced on our results.

In a study comparing PD patients with directional stimulation and omnidirectional stimulation, patients with directional stimulation had a 13% greater decline in daily anti-Parkinsonian medication than those with omnidirectional stimulation [18]. In our study, patients with SSA stimulation and patients with TSA and/or omnipolar stimulation had comparable results in LEDD reduction and in the improvement of the UPDRS-III scores, non-motor symptoms, and PDQ-39, as reported earlier [17]. There were no statistical differences in the demographics, baseline LEDD, and UPDRS-III scores of the patients with SSA activation compared to those with TSA and/or omnipolar stimulation.

Early reports on directional stimulation raised the concern of faster battery depletion due to higher impedances of directional stimulation [14,33]. Reports published soon after suggested that directional stimulation required a 31–40% lower therapeutic current strength (TCS) and 26% lower total energy delivered (TEED) in subthalamic and/or VIM-DBS [5,24,34]. It has been suggested that if TCS is diminished by at least 26%, the battery life span can be prolonged [16,34]. IPG’s battery consumption in bilateral directional stimulation was similar to other types of stimulation in this study, even with higher impedance. This could be the result of the lower current used with SSA.

dDBS has not presented greater risk in DBS lead implantation [35]. In a two-year retrospective analysis of Medicare and Medicaid Services, it was noted that patients with dDBS systems were 36% less likely to undergo re-operation than patients with the traditional DBS system [36]. This was thought to be related to differences in acute reoperations (i.e., infection, hardware malfunction, and unclassified reasons) and intracranial lead-related complications. It was estimated in a simulation study that a lead malposition of 1 mm could be overcome with directional leads [37]. In our study, only one patient went through additional DBS lead replacement during the follow-up in this cohort. No operation-related hemorrhages were noted, although 11% of patients were on anticoagulants or antithrombotic medication. No severe DBS operation-related infections were observed. Stimulation-related adverse effects were tolerable.

A limitation of this study is its retrospective nature and, therefore, there is a lack of comprehensive data during the follow-up. This study is single centered with its restrictions. Yet, we feel that this study gives an overview on the use of directional stimulation in a real-life setting.

## 5. Conclusions

Directional stimulation is widely used in real-life clinical care, and it is preferred by patients when they are given the opportunity to choose. dDBS appears to be reliable and feasible in the treatment of motor and non-motor symptoms with minimal side effects in advanced PD.

## Figures and Tables

**Figure 1 jpm-12-01224-f001:**
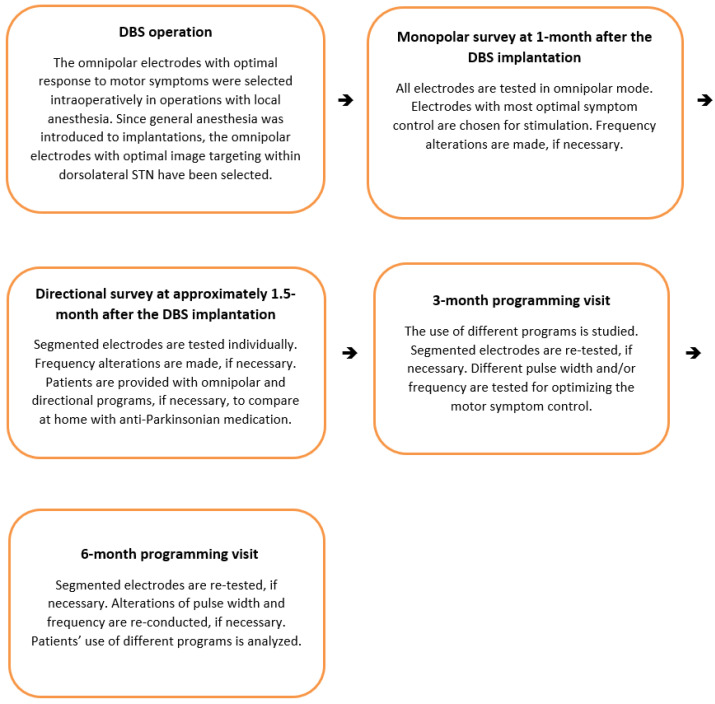
In-hospital deep brain stimulation (DBS) programming protocol.

**Figure 2 jpm-12-01224-f002:**
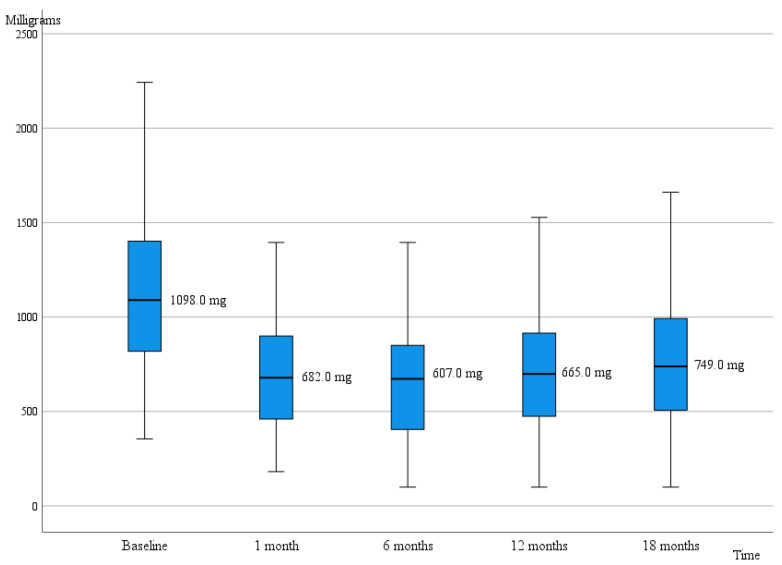
Median levodopa equivalent daily dosage, LEDD during the follow-up.

**Figure 3 jpm-12-01224-f003:**
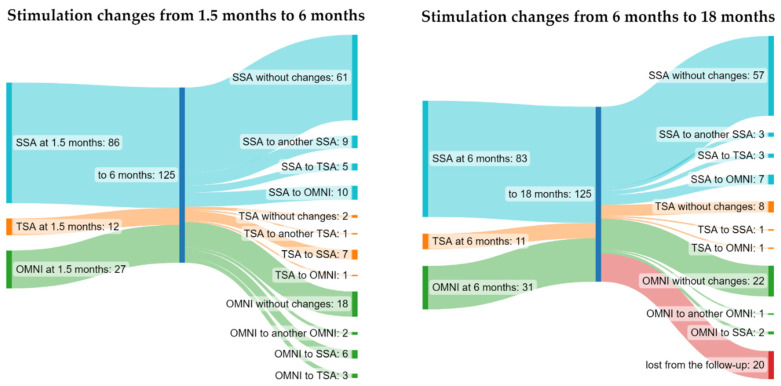
DBS stimulation changes during the 18-month follow-up. Numbers in each stimulation type represent the number of dDBS leads activated in each mode (SSA = single segment activation, TSA = two segment activation, OMNI = omnipolar stimulation, SSA to another SSA = SSA was changed within same electrode level, TSA to another TSA = TSA was changed within the same electrode level, OMNI to another OMNI = change in the active electrode level while stimulation was kept in omnipolar mode).

**Figure 4 jpm-12-01224-f004:**
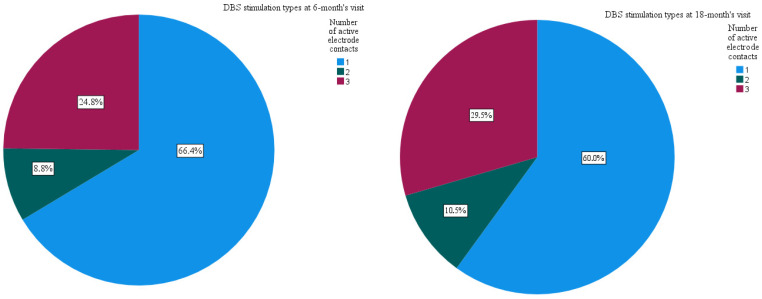
DBS stimulation types in dDBS leads at the 6-month visit and 18-month visit (1 = single segment activation, 2 = two segment activation, 3 = three segment activation/omnipolar stimulation).

**Table 1 jpm-12-01224-t001:** The demographics of the patients.

	Mean (SD)
Gender	35 (56%) male/28 (44%) female
Age	61.6 yrs (1.1 yrs)
Time from PD diagnosis to the DBS operation	11.1 yrs (3.9 yrs)
Time from the patient reported onset of motor fluctuations and/or dyskinesia to the DBS operation	3.0 yrs (0.3 yrs)
Comorbidities	64% know comorbidity 4% hypertension 5% atrial fibrillation 3% coronary artery disease 2% DM II
Concurrent medication	6% antithrombotic drug 5% anticoagulant 5% antidepressant

**Table 2 jpm-12-01224-t002:** Clinical results after the in-hospital six-month follow-up.

	Baseline, Median (IQR)	6-Month Programming Visit, Median (IQR)	
UPDRS part III, medication OFF state	34.6	20.7 (Stimulation ON)	*p* < 0.001 *
UPDRS part III, medication on	15.0 (8.0)		
LEDD	1098.0 (660.0)	673.0 (540.0)	*p* < 0.001 *
AIMS	10.0 (11.0)	1.0 (2.0)	*p* < 0.001 *
Hoehn and Yahr stage	3.0 (0.5)	2.0 (0.5)	*p* < 0.001 *
NMS-Quest	9.0 (5.0)	7.5 (7.0)	*p* = 0.021 *
PDQ-SI	23.8 (16.6)	24.4 (18.5)	*p* = 0.652
BDI	7.0 (5.0)	7.0 (8.0)	*p* = 0.860
MMSE	28.0 (3.0)	29.0 (4.0)	*p* = 0.032 *

* *p*-values under 0.05 were considered significant. UPDRS part III = Unified Parkinson’s Disease Rating Scale part 3, LEDD = levodopa equivalent daily dosage, AIMS = Abnormal Involuntary Movement Scale, NMS-Quest = Non-Motor Symptoms Questionnaire, PDQ-SI = Parkinson Disease Questionnaire 39 Summary Index, BDI = Beck Depression Inventory, MMSE = Mini-Mental Status Examination.

**Table 3 jpm-12-01224-t003:** DBS settings during the follow-up.

	1-Month Visit Median, (IQR)	6-Month Visit, Median (IQR)	12-Month Visit, Median (IQR)	18-Month Visit, Median (IQR)
**All stimulation types**				
**Current, mA**	1.4 (0.5)	2.2 (1.1)	2.5 (1.4)	2.7 (1.4)
**Pulse width, µs**	60.0 (0.0)	60.0 (0.0)	60.0 (0.0)	60.0 (0.0)
**Frequency, Hz**	130.0 (0.0)	130.0 (30.0)	130.0 (30.0)	130.0 (30.0)
**Impedance, Ω**	737.0 (182.0)	1693.5 (872.0)	1437.0 (762.0)	1350.0 (712.0)
**SSA**				
**Current, mA**	2.1 (1.3) ^¶^	2.2 (1.0)	2.3 (1.0)	2.2 (1.2)
**Pulse width, µs**	60.0 (0.0) ^¶^	60.0 (0.0)	60.0 (0.0)	60.0 (0.0)
**Frequency, Hz**	130.0 (0.0) ^¶^	130.0 (30.0)	130.0 (30.0)	130.0 (30.0)
**Impedance, Ω**	2099.0 (597.0) ^¶^	1950.0 (538.0)	1725.0 (563)	1637.0 (625.0)
**TSA/Omnipolar**				
**Current, mA**	2.4 (0.5) ^¶^	2.6 (1.4)	3.1 (2.0)	3.1 (1.7)
**Pulse width, µs**	60.0 (0.0) ^¶^	60.0 (0.0)	60.0 (0.0)	60.0 (0.0)
**Frequency, Hz**	130.0 (0.0) ^¶^	130.0 (15.0)	130.0 (38.0)	130.0 (43.0)
**Impedance, Ω**	1387.0 (538) ^¶^	1100.0 (305.0)	1062.5 (431.0)	994.0 (325.0)

^¶^ Data from the 1.5-month directional survey.

## Data Availability

The data is available by a request from the authors.

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
