# Peer review of "Real-Life Experience on Directional Deep Brain Stimulation in Patients with Advanced Parkinson’s Disease"

_jpm, 2022, doi:10.3390/jpm12081224_

Round 1

Reviewer 1 Report

The main aim of this study is to examine the real-life use of dDBS in advanced PD 40 patients treated with STN dDBS. The current study provides a long-term follow-up and provides a significant contribution to the literature. I recommend accepting it as it is. 

Author Response

We thank the reviewer for the review of the manuscript. We have done comprehensive style and English language editing for the manuscript.

Reviewer 2 Report

This is an informative analysis that presents longer follow-up periods compared to previous studies. The outcomes also align with the earlier studies with shorter follow-up periods. I recommend the following changes to the manuscript:

1.   Please spell out the acronyms (e.g., LEDD, VIM, H&Y, etc.) when they first appear in the article. Include a short description of what these terms mean as early as possible.  

2.   I recommend adding more background and content in the Introduction section. Information can be added to help the audience understand the following: how is Parkinson’s Disease generally managed with options including medications and DBS? In what cases is DBS adopted? What is the electrophysiology basis of the stimulation? What is our current knowledge of the mechanisms by which it alleviates symptoms? What is the difference between dDBS and traditional DBS from a setup standpoint?

3.   Recording the DBS stimulation changes over the follow-up period is informative. However, Fig. 3 is confusing and even inaccurate. The phrase “to 6-month follow-up: 125” in the blue box can make it seem like the number is SSA specific, although 125 is the total number of leads. Also, why does the color for OMNI change from left to right while the color for SSA and TSA stays the same? In the 1.5 month to 6 month diagram, the number of SSA doesn’t add up from left to right. On the left it is 86. On the right it is 85 (59+12+5+9). Is there patient dropout, or is it miscalculation/error in recording data? Similar inconsistencies occur in the 6 to 18 months diagram as well.  

4.   What is the rationale for stimulation changes? Were there observable characteristics of the patients that led to the physicians making changes in the stimulation? Can the authors name a few examples by listing the reasons for change in each scenario? Such information can be gathered from medical records or interviews with physicians.

5.  The manuscript is generally understandable but still requires extensive English editing to ensure correct grammar. For example, in line 15, 43% decline in LEDD was overserved” is more appropriate. In line 19, the preposition “to” is needed after “appears”.

6.   In the Discussion section, can the authors discuss why dDBS may be more favorable than conventional DBS?

Author Response

We thank the reviewer for the review. We have done comprehensive style and English language editing for the manuscript and re-done Figure 3 with alterations.

Please see the attachment for more detailed response.   
